# Continual Learning with Node-Importance based Adaptive Group Sparse Regularization

**Sangwon Jung**[1]*, **Hongjoon Ahn**[2]*, **Sungmin Cha**[1] **and Taesup Moon**[1,2]
[1]Department of Electrical and Computer Engineering, [2] Department of Artificial Intelligence,
Sungkyunkwan University, Suwon, Korea 16419
{s.jung, hong0805, csm9493, tsmoon}@skku.edu

## Abstract

We propose a novel regularization-based continual learning method, dubbed as Adaptive Group Sparsity based Continual Learning (AGS-CL), using two group sparsity-based penalties. Our method selectively employs the two penalties when learning each neural network *node* based on its the importance, which is adaptively updated after learning each task. By utilizing the proximal gradient descent method, the exact sparsity and freezing of the model is guaranteed during the learning process, and thus, the learner explicitly controls the model capacity. Furthermore, as a critical detail, we re-initialize the weights associated with unimportant nodes after learning each task in order to facilitate efficient learning and prevent the negative transfer. Throughout the extensive experimental results, we show that our AGS-CL uses orders of magnitude less memory space for storing the regularization parameters, and it significantly outperforms several state-of-the-art baselines on representative benchmarks for both supervised and reinforcement learning.

## 1 Introduction

Continual learning, also referred to as lifelong learning, is a long standing open problem in machine learning, in which the training data is given sequentially in a form divided into the groups of tasks. The goal of continual learning is to overcome the fundamental trade-off: the *stability-plasticity dilemma* [7, 21], *i.e.*, if the model focuses too much on the stability, it suffers from poor forward transfer to the new task, and if it focuses too much on the plasticity, it suffers from the catastrophic forgetting of past tasks. To address this dilemma, a comprehensive study for neural network-based continual learning was conducted broadly under the following categories: regularization-based [18, 13, 41, 22, 1, 3], dynamic architecture-based [27, 39, 10], and replay memory-based [26, 20, 34, 11] methods.

In this paper, we focus on the regularization-based methods, since they pursue to use the fixed-capacity neural network model as efficiently as possible, which may potentially allow them to be combined with other approaches. These methods typically identify important learned *weights* for previous tasks and heavily penalize their deviations while learning new tasks. They have a natural connection with a separate line of research, the model compression of neural networks [17, 19, 42]. Namely, in order to obtain a compact model, typical model compression methods measure the importance of each node or weight in a given neural network and prune the unimportant ones, hence, share the similar principle with the regularization-based continual learning schemes. Several representative model compression methods [37, 4, 38, 29] used the group Lasso-like penalties, which define the incoming or outgoing weights to a *node* as groups and achieve structured sparsity within a neural network. Such focus on the node-level importance could lead to a more efficient representation of the model and achieved better compression than focusing on the weight-wise importance.

---

Inspired by such connection, we propose a new regularization-based continual learning method, dubbed as Adaptive Group Sparsity based Continual Learning (AGS-CL), that can adaptively control the plasticity and stability of a neural network learner by using two *node-wise* group sparsity-based penalties as regularization terms. Namely, our first term, which is equivalent to the ordinary group Lasso penalty, assigns and learns new important nodes when learning a new task while maintaining the structured sparsity (*i.e.*, controls plasticity), whereas the second term, which is a group sparsity penalty imposed on the *drifts* of the important node parameters, prevents the forgetting of the previously learned important nodes via freezing the incoming weights to the nodes (*i.e.*, controls stability). The two terms are selectively applied to each node based on the *adaptive* regularization parameter that represents the importance of each node, which is updated after learning each new task. For learning, we utilize the proximal gradient descent (PGD) [23] such that the *exact* sparsity and freezing of the nodes can be elegantly attained, without any additional threshold to tune. Moreover, as a critical detail, we re-initialize the weights associated with the *unimportant* nodes after learning each task, such that the negative transfer can be prevented and plasticity can be maximized.

As a result, we convincingly show our AGS-CL efficiently mitigates the catastrophic forgetting while continuously learning new tasks, throughout extensive experiments on several benchmarks in *both* supervised and reinforcement learning. Our experimental contributions are multifold. First, we show that AGS-CL significantly outperforms strong state-of-the-art baselines [13, 41, 2, 8] on *all* of benchmark datasets we tested. Second, we give a detailed analysis on the stability-plasticity trade-off of our model, by utilizing additional metrics beyond average accuracy. Third, we identify AGS-CL uses *orders of magnitude* less additional memory than the baselines to store the regularization parameters, thanks to only maintaining the node-wise regularization parameters. Such compact memory usage is a nice by-product and enables applying our method to much larger networks, which typically is necessary for applications with large-scale datasets. Finally, we stress that our RL results on Atari games are for the *pure* continual learning setting, in which past tasks cannot be learned again, in contrast to other works [13, 31] that allow the agents to learn multiple tasks in a *recurring* fashion.

**Related work** Diverse approaches for neural networks based continual learning have been proposed, as exhaustively surveyed in [24]. Unlike the typical weight-wise regularization-based methods, *e.g.*, [13, 41, 8, 2, 22], several other works considered the node-wise importance, similarly as ours, as well, but had some limitations. For instance, [1] considered node-importance in the context of Bayesian neural network and variational inference, but their method had to work with several heuristic-based losses and cannot be applied to non-Bayesian pre-trained models. [33] utilized a node-wise hard attention mechanism per layer to freeze the important nodes, but they required to know the total number of tasks to be learned in advance and had to implement a subtle annealing heuristic for attention. [3] devised additional regularization term for promoting node-wise sparsity to boost the performance of the weight-wise regularization based methods, but the scheme still had to store the weight-wise regularization parameters. [10] developed a notion of active and inactive nodes and implemented pruning and freezing schemes, which are similar to ours, but they required several hyperparameters and involved cumbersome, non-adaptive threshold tuning steps that require a separate validation set. In contrast, our AGS-CL employs more principled loss function and optimization routine, unlike [1, 33, 10], stores only node-wise regularization parameters, unlike [3], and automatically determines which nodes to prune or freeze, unlike [10].

The group Lasso [40] regularization, which was favorably used in model compression [4, 37, 38, 29], has been also adopted for continual learning in [39]. However, they considered a setting in which model capacity can grow as the learning continues, which is different from our focus, and their method involved many hyperparameters and multiple re-training steps, which make it hard to apply in practice. Moreover, there was no mechanism to freeze the model at the group level in [39].

## 2 Motivation

Here, we give the main motivation for our algorithm. We start from the intuition that the node in a neural network is the basic unit for representing the learned information from a task, and the catastrophic forgetting occurs when the information flowing to the important nodes changes as the learning continues with new tasks. Namely, assume the important nodes for task $t - 1$ are identified. Then, we argue that there are two sources for the catastrophic forgetting: *model drift* and *negative transfer* as shown in Figure 1. The model drift corresponds to the case in which the incoming weights of an important node (node $j$ in Figure 1) gets changed when learning a new task $t$. In this case,

the representation of node $j$ for task $t-1$ can alter, hence, the performance for task $t-1$ can drastically degrade. The red arrows and dotted lines in the figure exemplify such model drift for node $j$. On the other hand, the negative transfer happens when the representation of an unimportant node for task $t-1$ in the lower layer (node $i$ in Figure 1) changes during learning a new task $t$. Namely, even when there is no model drift, if node $i$ becomes important for task $t$, then such change of representation will bring an interfering effect for node $j$ when carrying out task $t-1$. The color change of $i$ and red arrow in the figure shows such negative transfer from the future tasks.

In order to address above two key sources of catastrophic forgetting, we believe an effective continual learning algorithm should essentially carry out the followings:

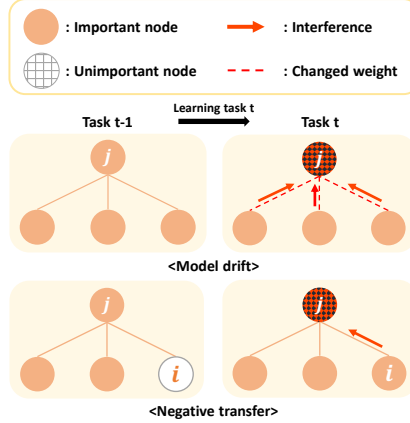

- *Freeze* important nodes: Once a node has been identified as important, its *incoming* weights should be frozen while learning future tasks, hence, the model drift can be prevented.
- *Nullify* transfer from unimportant nodes: Once a node has been identified as unimportant, its *outgoing* weights should be fixed to $0$ (*i.e.*, pruned), hence, the negative transfer from the node to the upper layers can be eliminated.

We note most of the state-of-the-art regularization-based methods aim to approximate the first item via regularizing the important weights, while largely neglecting the second item. One exception is [10], but as mentioned in related work, their method required multiple heuristics to determine unimportant nodes and prune the outgoing weights of the unimportant nodes. Our proposed AGS-CL, on the other hand, automatically determines the important and

Figure 1: Two sources of catastrophic forgetting: model drift (top) and negative transfer (bottom).

unimportant nodes as the learning continues, freezes the incoming weights for the important ones, and nullifies the outgoing weights for the unimportant ones, all via selectively applying two group sparsity based penalties based on the adaptive regularization parameter defined for each node. Furthermore, to maximize the plasticity, the random initialization of the *incoming* weights of unimportant nodes are implemented as well, and we elaborate each step more in details in the next section.

## 3 Adaptive Group Sparsity based Continual Learning (AGS-CL)

### 3.1 Notations

We denote $\ell \in \{1 \ldots, L\}$ as a layer of a neural network model that has $N_\ell$ nodes, and let $n_\ell \in \{1, \ldots, N_\ell\}$ be a node in that layer. For the convolutional neural networks (CNN), a node stands for a convolution filter (or channel). Moreover, $\boldsymbol{\theta}_{n_\ell}$ denotes the vector of the (incoming) weight parameters for the $n_\ell$-th node. Hence, $\theta_{n_\ell,i}$ stands for the weight that connects the $i$-th node (or channel) in layer $\ell-1$ with the node $n_\ell$. Moreover, $\mathcal{G} \triangleq \{n_\ell : 1 \leq n_\ell \leq N_\ell, 1 \leq \ell \leq L\}$ is the set of all the nodes in the neural network, and $\boldsymbol{\theta} = \{\boldsymbol{\theta}_{n_\ell}\}_{n_\ell \in \mathcal{G}}$ denotes the entire parameter vector of the network. We assume ReLU is always used as the activation function for all layers. We denote $\mathcal{D}_t$ as the training dataset for task $t \in \{1, \ldots, \mathcal{T}\}$, and we assume the task boundaries are given to the learner.

### 3.2 Loss function

Before describing the loss function for task $t$, we first introduce the adaptive regularization parameter $\Omega_{n_\ell}^{t-1} \geq 0$ for each node $n_\ell \in \mathcal{G}$, of which magnitude indicates how important the node is for carrying out the tasks up to $t-1$. Namely, large $\Omega_{n_\ell}^{t-1}$ indicates that the node $n_\ell$ has been identified and learned as important, and $\Omega_{n_\ell}^{t-1} = 0$ denotes the node $n_\ell$ was not important for learning tasks up to $t-1$. The exact definition and update mechanism for $\Omega_{n_\ell}^{t-1}$ are given in Section 3.4, but for now, we assume such parameter is given when learning a new task $t$.

Given such $\{\Omega_{n_\ell}^{t-1}\}_{n_\ell \in \mathcal{G}}$, we define a set of unimportant nodes as

$$\mathcal{G}_0^{t-1} \triangleq \{n_\ell : \Omega_{n_\ell}^{t-1} = 0\} \subseteq \mathcal{G}, \tag{1}$$

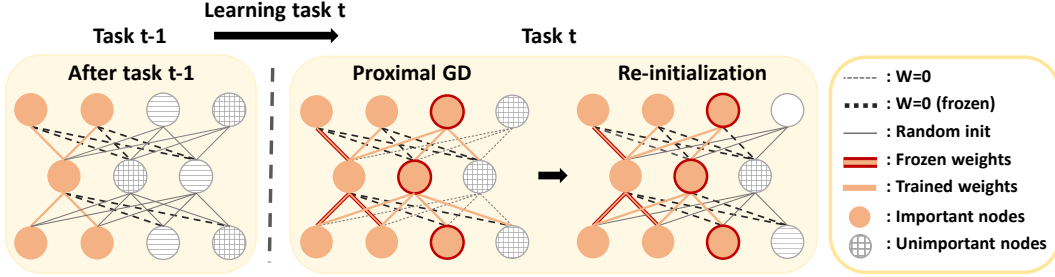

Figure 2: Summary of AGS-CL. During learning a new task $t$, the PGD step using term ($a$) in (2) identifies the *new* important nodes (orange, solid nodes with red boundaries) and remaining unimportant nodes (gray, checkered nodes). The incoming weights connected to the *sufficiently* important nodes up to $t - 1$ (vanilla orange, solid nodes) are frozen at $t$ (orange lines with red boundaries) due to the PGD step with term ($b$) in (2). The re-initialization step then fixes the outgoing weights of unimportant nodes to zero (black, bold dotted lines) and randomly initializes the incoming weights of unimportant nodes (gray, solid thin weights).

and with the training data $\mathcal{D}_t$, our loss function for learning task $t$ is defined as

$$\mathcal{L}_t(\boldsymbol{\theta}) = \mathcal{L}_{\text{TS},t}(\boldsymbol{\theta}) + \mu \underbrace{\sum_{n_\ell \in \mathcal{G}_0^{t-1}} \|\boldsymbol{\theta}_{n_\ell}\|_2}_{(a)} + \lambda \underbrace{\sum_{n_\ell \in \mathcal{G} \setminus \mathcal{G}_0^{t-1}} \Omega_{n_\ell}^{t-1} \|\boldsymbol{\theta}_{n_\ell} - \hat{\boldsymbol{\theta}}_{n_\ell}^{(t-1)}\|_2}_{(b)} . \tag{2}$$

In (2), $\mathcal{L}_{\text{TS},t}(\boldsymbol{\theta})$ stands for the ordinary task-specific loss on $\mathcal{D}_t$ (*e.g.*, cross-entropy for supervised learning), and the terms ($a$) and ($b$) are the group sparsity-based regularization terms, in which $\hat{\boldsymbol{\theta}}_{n_\ell}^{(t-1)}$ is the learned parameter vector for node $n_\ell$ up to task $t - 1$, and $\mu, \lambda \geq 0$ are the hyperparameters that set the trade-offs among the penalty terms. Notice that our loss function *selectively* employs the regularization terms based on the value of $\Omega_{n_\ell}^{t-1}$. Namely, for the unimportant nodes in $\mathcal{G}_0^{t-1}$, we apply the group Lasso penalty (term ($a$)) as in [4], and for the important nodes in $\mathcal{G} \setminus \mathcal{G}_0^{t-1}$, we apply the group-sparsity based deviation penalty (term ($b$)) that adaptively penalizes (similarly as in [43, 35]) the deviation of $\boldsymbol{\theta}_{n_\ell}$ from $\hat{\boldsymbol{\theta}}_{n_\ell}^{(t-1)}$ depending on the magnitude of $\Omega_{n_\ell}^{t-1} > 0$.

We elaborate that term ($a$) controls the *plasticity* of the model when learning new tasks, whereas term ($b$) is in charge of achieving the *stability* via preventing the model drift mentioned in Section 2. Namely, term ($a$) automatically identifies the active learners for the new task $t$ among the unimportant nodes so far and sparsifies the rest of the nodes such that they can be allocated for learning future tasks. On the other hand, the term ($b$) enforces to freeze a node (*i.e.*, prevent model drift) if it has been identified to be important enough, *i.e.*, it has large $\Omega_{n_\ell}^{t-1}$ value. Note due to the property of the group-norm penalties, the sparsification and freezing resulting from applying the two regularization terms can be *exact* when appropriate optimization method is used, as described in the next subsection.

## 3.3 Learning with proximal gradient descent

While directly minimizing $\mathcal{L}_t(\boldsymbol{\theta})$ can be done via applying vanilla SGD-variant optimizers, *e.g.*, Adam [12], we employ the proximal gradient descent (PGD) method [23, Section 4.2]. To that end, we first denote the proximal operator as

$$\textbf{prox}_{\alpha f}(\boldsymbol{v}) = \arg \min_{\boldsymbol{\theta}} \left( f(\boldsymbol{\theta}) + \frac{1}{2\alpha} \|\boldsymbol{\theta} - \boldsymbol{v}\|_2^2 \right) \tag{3}$$

for a scalar $\alpha > 0$ and a convex function $f$. Then, by simply denoting (2) as $\mathcal{L}_t(\boldsymbol{\theta}) = \mathcal{L}_{\text{TS},t}(\boldsymbol{\theta}) + \mathcal{L}_{\text{Reg},t}(\boldsymbol{\theta})$, in which $\mathcal{L}_{\text{Reg},t}(\boldsymbol{\theta})$ is the *convex* regularization term that combines term ($a$) and term ($b$) in (2), the PGD with learning rate $\alpha$ iteratively minimizes (2) by computing the following

$$\tilde{\boldsymbol{\theta}}^{k+1} := \boldsymbol{\theta}^k - \alpha \nabla \mathcal{L}_{\text{TS},t}(\boldsymbol{\theta}^k) \tag{4}$$

$$\boldsymbol{\theta}^{k+1} := \textbf{prox}_{\alpha \mathcal{L}_{\text{Reg},t}(\boldsymbol{\theta})} \left( \tilde{\boldsymbol{\theta}}^{k+1} \right). \tag{5}$$

for $k = 0, \ldots, \mathcal{K} - 1$. Namely, $\boldsymbol{\theta}^k$ is the $k$-th proximal gradient update step. Namely, (5) applies the proximal operator (3) with $f = \mathcal{L}_{\text{Reg},t}(\boldsymbol{\theta})$ on the gradient update of $\boldsymbol{\theta}^k$ using $\nabla \mathcal{L}_{\text{TS}}(\boldsymbol{\theta}^k)$. Now, for deriving a succinct, concrete parameter update rule for our algorithm, we introduce the following lemma, of which proof is given in the Supplementary Material.

**Lemma 1** *For $f(\boldsymbol{\theta}) = c\|\boldsymbol{\theta} - \boldsymbol{\theta}_0\|_2$ with $c > 0$ and any fixed vector $\boldsymbol{\theta}_0$,*

$$\mathbf{prox}_{\alpha f}(\boldsymbol{v}) = \gamma \boldsymbol{v} + (1 - \gamma)\boldsymbol{\theta}_0, \tag{6}$$

*in which $\gamma = \left(1 - \frac{\alpha c}{\|\boldsymbol{\theta}_0 - \boldsymbol{v}\|_2}\right)_+$ where $(x)_+ = \max\{0, x\}$.*

From (3), we can easily see that the proximal operator can be applied to each node parameter vector $\boldsymbol{\theta}_{n_\ell}$, or each group, independently when carrying out (5). Hence, by Lemma 1, we have the following closed-form proximal gradient update rules:

$$\boldsymbol{\theta}_{n_\ell}^{k+1} = \begin{cases} \left(1 - \frac{\alpha\mu}{\|\tilde{\boldsymbol{\theta}}_{n_\ell}^{k+1}\|_2}\right)_+ \tilde{\boldsymbol{\theta}}_{n_\ell}^{k+1} & \text{for} \quad n_\ell \in \mathcal{G}_0^{t-1} \\ \gamma\tilde{\boldsymbol{\theta}}_{n_\ell}^{k+1} + (1 - \gamma)\hat{\boldsymbol{\theta}}_{n_\ell}^{(t-1)} & \text{for} \quad n_\ell \in \mathcal{G}\backslash\mathcal{G}_0^{t-1}, \end{cases} \tag{7}$$

in which $\gamma = \left(1 - \frac{\alpha\lambda\Omega_{n_\ell}^{t-1}}{\|\tilde{\boldsymbol{\theta}}_{n_\ell}^{k+1} - \hat{\boldsymbol{\theta}}_{n_\ell}^{(t-1)}\|_2}\right)_+$. Note the first rule in (7) can set $\boldsymbol{\theta}_{n_\ell}^{k+1} = \mathbf{0}$ (*i.e.*, sparsify) when $\|\tilde{\boldsymbol{\theta}}_{n_\ell}^{k+1}\|_2 \leq \alpha\mu$ for the unimportant nodes, and the second rule can set $\boldsymbol{\theta}_{n_\ell}^{k+1} = \hat{\boldsymbol{\theta}}_{n_\ell}^{(t-1)}$ (*i.e.*, freeze) when $\|\tilde{\boldsymbol{\theta}}_{n_\ell}^{k+1} - \hat{\boldsymbol{\theta}}_{n_\ell}^{(t-1)}\|_2 \leq \alpha\lambda\Omega_{n_\ell}^{t-1}$ for the important nodes. Thus, we can automatically achieve the *exact* sparsification and freezing of the node parameters as a part of the optimization routine, *without* any additional thresholds or heuristics that are otherwise required when using vanilla SGD-variants or [10]. Moreover, we show in the Supplementary Material that the accurate sparsification and freezing by our PGD update is integral in achieving high accuracy by comparing with a scheme without it. Finally, from the theory of PGD [23], (7) is guaranteed to converge to a local minima of $\mathcal{L}_t(\boldsymbol{\theta})$ with appropriate $\alpha$. The converged parameters are then denoted as $\hat{\boldsymbol{\theta}}^{(t)} = \{\hat{\boldsymbol{\theta}}_{n_\ell}^{(t)}\}_{n_\ell \in \mathcal{G}}$.

### 3.4 Updating $\Omega_{n_\ell}^{t-1}$ and re-initialization of unimportant nodes

**Updating $\Omega_{n_\ell}^{t-1}$** Now, we give the definition and the update formula of $\{\Omega_{n_\ell}^{t-1}\}_{n_\ell \in \mathcal{G}}$, which reflect the importance of nodes and play a crucial role in our loss function. Initially, we set $\Omega_{n_\ell}^0 = 0$ for all $n_\ell \in \mathcal{G}$, thus, for $t = 1$, we obtain the ordinary group Lasso solution since $\mathcal{G}_0^0 = \mathcal{G}$. After minimizing $\mathcal{L}_t(\boldsymbol{\theta})$, $\Omega_{n_\ell}^t$ gets updated as

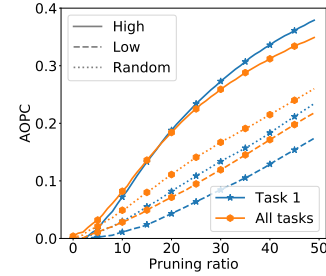

$$\Omega_{n_\ell}^t := \eta\Omega_{n_\ell}^{t-1} + \frac{1}{N_t}\sum_{m=1}^{N_t} a_{n_\ell}(\boldsymbol{x}_m^{(t)}) \tag{8}$$

for all $n_\ell \in \mathcal{G}$, in which $a_{n_\ell}(\boldsymbol{x}_m^{(t)})$ is the ReLU activation value of the node $n_\ell$ when the input data is $\boldsymbol{x}_m^{(t)} \in \mathcal{D}_t$, and $\eta \in (0, 1]$ is the hyperparameter for the exponential averaging. Hence, we regard the average activation value of $n_\ell$ for task $t$ as the *importance* of the node, and it is added to $\Omega_{n_\ell}^{t-1}$. Namely, a node remains unimportant (*i.e.*, be in $\mathcal{G}_0^t$) when either the incoming weights remain to be zero

Figure 3: AOPC for $\{\Omega_{n_\ell}^t\}$

or the ReLU activations are dead for all training data points, after learning task $t$. Furthermore, $\eta < 1$ implements exponential moving average, similarly as in [31], such that the $\{\Omega_{n_\ell}^t\}$ values do not explode (we always used $\eta = 0.9$).

One may argue whether the average ReLU activation as in (8) can be a correct measure for identifying the importance of a node. To that end, Figure 3 justifies our choice by considering Area Over Prediction Curve (AOPC) [28] for $\{\Omega_{n_\ell}^t\}$ on CIFAR-100 [14] tasks, which splits 100 classes into 10 tasks. AOPC is a widely used metric for quantitatively evaluating the neural network interpretation methods, *e.g.*, [32, 5], and a steep increase of AOPC with respect to the pruning (or perturbing) of nodes (or pixels) in the order of high importance values suggests the validity of an interpretation method. Figure 3 shows AOPC curves of our importance measure $\{\Omega_{n_\ell}^t\}$, in which the pruning of nodes is done in the order of random (dotted), highest (solid) and lowest (dashed) values after

learning task 1 (blue line with star) and all tasks (orange line with circle), respectively. We clearly observe the significant gaps between the solid and dashed/dotted lines, which corroborates the validity of using average ReLU activation for $\{\Omega_{n_\ell}^t\}$. We note some alternatives for $\{\Omega_{n_\ell}^t\}$ may be also used, *e.g.*, apply neural network interpretation methods, but due to the simplicity (*i.e.*, only requiring forward-pass in contrast to [32, 5] that also require backward-pass) and correctness shown in Figure 3, we adhere to using (8) and defer to future work for comparing with other interpretation methods.

**Re-initialization:** Once $\{\Omega_{n_\ell}^t\}_{n_\ell \in \mathcal{G}}$ are updated, we carry out two re-initialization steps on the weights that are connected to the unimportant nodes in $\mathcal{G}_0^t$. That is, for the weights $\hat{\boldsymbol{\theta}}^{(t)} = \{\hat{\boldsymbol{\theta}}_{n_\ell}^{(t)}\}_{n_\ell \in \mathcal{G}}$,

    (I.1) **[Zero-init]** Fix $\hat{\theta}_{n_\ell,i}^{(t)} = 0$ if $i \in \mathcal{G}_0^t$, for all future tasks after $t$.

    (I.2) **[Rand-init]** Randomly initialize $\hat{\boldsymbol{\theta}}_{n_\ell}^{(t)}$ if $n_\ell \in \mathcal{G}_0^t$, with probability $\rho$.

The former fixes the *outgoing* weights of an unimportant node to zero (*i.e.*, prunes) for *all* remaining tasks, while the latter randomly initializes (*i.e.*, frees) the *incoming* weight vector of an unimportant node, with probability $\rho$. We can see (I.1) prevents the negative transfer mentioned in Section 2 and improves stability, since the change of the representations of unimportant nodes in $\mathcal{G}_0^t$ happening in the future tasks will *never* affect the important nodes for task $t$ in the upper layer that are connected to $i$. Moreover, we observe (I.2) enables some nodes in $\mathcal{G}_0^t$ to become *active learners* for future tasks and improves plasticity, since the incoming weights of those nodes would otherwise typically not get updated due to zero gradient. $\rho \in (0, 1]$ is a hyperparameter that controls the capacity of the network for learning new tasks, and $\rho \leq 0.5$ typically shows good trade-off between the sparsity and used capacity of the network.

We also emphasize that the order of the re-initialization steps is important, *i.e.*, (I.1) is always followed by (I.2), which can be seen by observing that the outgoing weights of an unimportant node can be connected to *either* important *or* unimportant nodes in the upper layer. Namely, in such a case, (I.1) *nullifies* and *fixes* those connected to the important nodes, but (I.2) re-utilizes those connected to the unimportant nodes so that they can become learnable again for the next task. This is also illustrated in the rightmost network figure in Figure 2; note the activation of the unimportant node in the second layer can be used for learning the unimportant node in the third layer for task $t + 1$ (via the gray, solid thin weight between them that is randomly re-initialized). In our experimental results, we systematically show the critical effects of (I.1) and (I.2).

Finally, we summarize our method in Algorithm 1 and Figure 2.

---

**Algorithm 1** AGS-CL algorithm

---

**Require:** $\{\mathcal{D}_t\}_{t=1}^{\mathcal{T}}$: Sequential training datasets
**Require:** $\mu, \lambda, \rho$: Hyperparamters, $\mathcal{K}$: Number of epochs for each task
  Randomly initialize $\boldsymbol{\theta}$ and set $\Omega_{n_\ell}^0 = 0, \forall n_\ell \in \mathcal{G}$.
  **for** $t = 1, \cdots, \mathcal{T}$ **do**
    Define the loss function $\mathcal{L}_t(\boldsymbol{\theta})$ in (2).
    **for** $k = 0, \cdots, \mathcal{K} - 1$ **do**
      Compute (4) and (5) together with (7) to obtain $\boldsymbol{\theta}_{n_\ell}^k$ for each $n_\ell \in \mathcal{G}$   `/*PGD updates*/`
    **end for**
    Obtain $\hat{\boldsymbol{\theta}}^{(t)}$ and update $\{\Omega_{n_\ell}^t\}_{n_\ell \in \mathcal{G}}$ using (8)   `/*Update` $\{\Omega_{n_\ell}^t\}$`*/`
    Obtain $\mathcal{G}_0^t$ as in (1)
    Re-initialize $\hat{\boldsymbol{\theta}}^{(t)}$ using first with (I.1), then with (I.2)   `/*Re-initializations*/`
  **end for**

---

# 4 Experimental Results

## 4.1 Supervised learning on vision datasets

We evaluate the performance of AGS-CL together with the representative regularization-based methods, EWC [13], SI [41], RWALK [8], MAS [2], and HAT [33]. We used multi-headed outputs for all experiments, and 5 different random seed runs (that also shuffle task sequences except for Omniglot) are averaged for all datasets. We tested on multiple different vision datasets and thoroughly showed the effectiveness of our method: CIFAR-10/100 [14] was used as a standard benchmark

with smaller number of tasks, Omniglot [16] was used to compare the performance for large number of tasks, CUB200 [36] was used to test on more complex, large-scale data, and the sequence of 8 different datasets, {CIFAR-10 / CIFAR-100 / MNIST / SVHN / Fashion-MNIST / Traffic-Signs / FaceScrub / NotMNIST}, which was proposed in [33], was used to test the check the learning capability for different visual domains.

For all the experiments, we used convolutional neural networks (CNN) with ReLU activations, of which architectures are the followings: for CIFAR-10/100, we used 6 convolution layers followed by 2 fully connected layers, for Omniglot, we used 4 convolution layers as in [31], for CUB200, we used AlexNet [15] pre-trained on ImageNet [9], and for the mixture of different tasks, we used AlexNet trained from scratch. We fairly searched the hyperparameters for all baselines and report the best performance for each method. Our method was implemented with PyTorch [25], and Adam [12] step was used as $\nabla \mathcal{L}_{\text{TS},t}(\boldsymbol{\theta}^k)$ in (4), and PGD update (5) was applied once after each epoch. More details and ablation studies on the hyperparameters (particularly for $\rho$ and PGD updates) as well as on model architectures with full hyperparameter settings are given in the Supplementary Material.

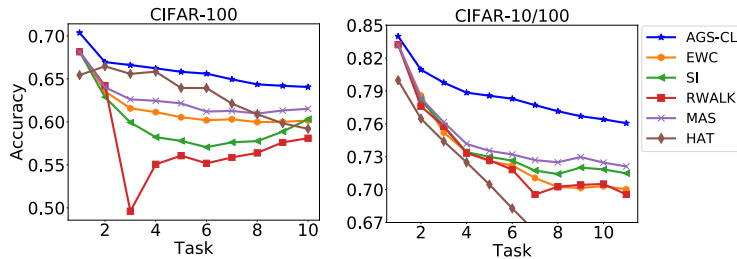

Figure 4: Average accuracy results on CIFAR-100 and CIFAR-10/100 datasets.

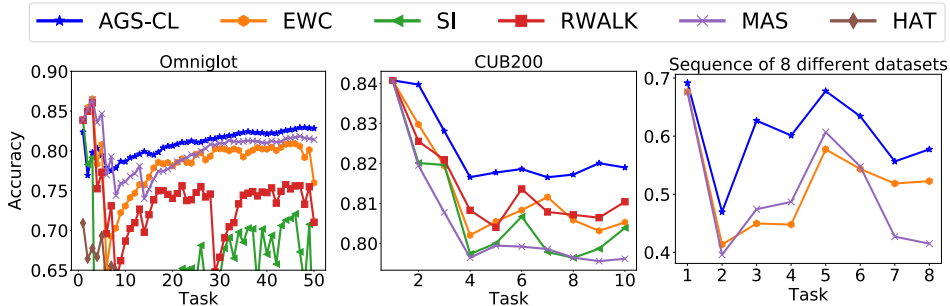

Figure 5: Average accuracy results on Omniglot, CUB200, and the sequence of 8 datasets.

**Average accuracy** Figure 4 and 5 show the average accuracy result on each dataset. The first figure in Figure 4 is on CIFAR-100, which splits 100 classes into 10 tasks with 10 classes per task, and the second is on CIFAR-10/100, which additionally uses CIFAR-10 for pre-training before learning tasks from CIFAR-100. In Figure 5, the first figure is on Omniglot, which treats each alphabet as a single task and uses all 50 alphabets, the second figure is on CUB200, which splits 200 classes into 10 tasks with 20 classes per task, and the third figure is on the sequence of 8 different vision datasets, which treats each dataset as a separate task. For the first and third figures, there were different numbers of classes for each task, and the total number of classes was 1600 and 293. For the sequence of 8 different datasets, we only compared with EWC and MAS since they were the two best baselines on other datasets. We can make the following observations from the results. Firstly, we clearly observe that our AGS-CL consistently dominates other baselines for all the datasets throughout most tasks. We stress that this is significant since AGS-CL uses much smaller memory to store the regularization parameters than others, as more elaborated below. Secondly, among other baselines, there is no clear winner; MAS tends to excel in the first three sets, while it is the worst for CUB200 and the 8 different vision datasets. Thirdly, as seen in the results for Omniglot, SI and RWALK, which are based on path integral of gradient vector field, had large performance variance for larger number of tasks.

**Analysis 1: Required memory size** Figure 6(a) compares the required memory sizes to store regularization parameters between AGS-CL and other weight-wise regularization methods (*e.g.*, MAS). Note AGS-CL only needs to store the parameters for the nodes, $\{\Omega_{n_\ell}^t\}$, whereas others need

to store the parameters for the weights. From the figure, we clearly observe that AGS-CL uses *orders of magnitude* less memory than other methods. Particularly, for CUB200, in which a large-scale model (AlexNet) is used, the gap is more than four orders, and such drastic compactness in additional memory gives a competitive edge on the practicality of our AGS-CL.

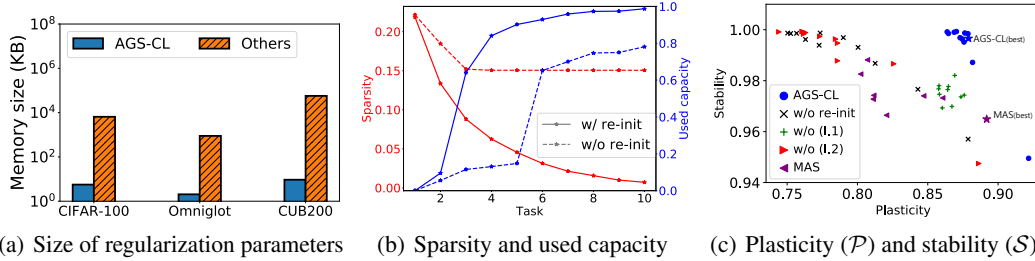

(a) Size of regularization parameters     (b) Sparsity and used capacity     (c) Plasticity ($\mathcal{P}$) and stability ($\mathcal{S}$)

Figure 6: Various analyses on AGS-CL. (b) and (c) are for CIFAR-100. We note the decreasing and increasing curves in (b) represent sparsity and used capacity, respectively.

**Analysis 2: Sparsity and used capacity of the model** Figure 6(b) closely analyzes how sparsity, the left (red) $y$-axis, and used capacity, the right (blue) $y$-axis, of the network evolve as the learning with AGS-CL continues, for CIFAR-100. Moreover, the solid and dashed lines represent the schemes with or without the re-initialization steps, *i.e.*, (I.1) and (I.2), respectively, and we set $\rho = 0.3$ for (I.2). We define the sparsity and the used capacity of a network as the ratios

$$\frac{|\mathcal{G}_0^t|}{|\mathcal{G}|} \quad \text{and} \quad \frac{|\{n_\ell : \|\hat{\boldsymbol{\theta}}_{n_\ell}^{(t)} - \hat{\boldsymbol{\theta}}_{n_\ell}^{(t-1)}\|_2 = 0\}|}{|\mathcal{G}|} ,$$

respectively. Thus, large sparsity implies many "active learners" are available for learning future tasks, and large used capacity means many nodes are frozen to not forget past tasks. We observe the sparsity and the used capacity gradually decreases and increases, respectively, automatically controlled by $\{\Omega_{n_\ell}^t\}$ and PGD as intended. We further observe that the re-initialization steps are essential for AGS-CL; without the re-initialization, the network sparsity does not drop beyond a certain level, hence, AGS-CL cannot utilize the full capacity of the network.

**Analysis 3: Effect of re-initializations** To further study the effectiveness of re-initialization more concretely, we evaluated two additional metrics, plasticity ($\mathcal{P}$) and stability ($\mathcal{S}$). To define the metrics, we first let $A \in \mathbb{R}^{\mathcal{T} \times \mathcal{T}}$ be the accuracy matrix of a continual learning algorithm, in which $A_{ij}$ is the accuracy of the $j$-th task after learning the $i$-th task, and let $A_i^*$ be the accuracy of a vanilla fine-tuning scheme for task $i$. Then, the metrics are defined as

$$\mathcal{P} \triangleq \frac{1}{\mathcal{T}} \sum_{i=1}^{\mathcal{T}} \frac{A_{ii}}{A_i^*} \quad \text{and} \quad \mathcal{S} \triangleq \frac{1}{\mathcal{T}} \sum_{j=1}^{\mathcal{T}} \frac{A_{\mathcal{T}j}}{\max_{j \leq i \leq \mathcal{T}}(A_{ij})} ,$$

in which $\mathcal{P}$ measures the amount of "forward transfer" and $\mathcal{S}$ measures the amount of "not forgetting" (*i.e.*, higher the better for both). Figure 6(c) reports the trade-offs between $\mathcal{P}$ and $\mathcal{S}$, obtained from CIFAR-100 for several variants of AGS-CL and a representative baseline, MAS. For AGS-CL, we ablated each re-initialization scheme in Section 3.4; '*w/o* (I.1)' is without (I.1) step, '*w/o* (I.2)' is without (I.2) step, and '*w/o* re-init' is without both. Moreover, the plotted trade-offs are over the hyperparameters; *i.e.*, for AGS-CL, we fixed $(\mu, \rho) = (10, 0.3)$ and varied $\lambda$, and for MAS, we varied $\lambda$. The two '$\star$' points in the figure represent the results of the optimum $\lambda$ for AGS-CL (blue) and MAS (purple). Followings are our observations. First, we clearly see AGS-CL has much better $\mathcal{P}$-$\mathcal{S}$ trade-off than MAS. Namely, AGS-CL hardly suffers from any forgetting (*i.e.*, $\mathcal{S} \approx 1$) and has higher $\mathcal{P}$ values than MAS for most cases. Second, we clearly observe (I.1) improves stability, by comparing AGS-CL and '*w/o* (I.1)' at similar $\mathcal{P}$, and (I.2) improves plasticity, by comparing AGS-CL and '*w/o* (I.2)' at similar $\mathcal{S}$. Finally, '*w/o* re-init' and '*w/o* (I.2)' show similar performance, hence, (I.1) alone is not enough for attaining both high $\mathcal{P}$ and $\mathcal{S}$.

## 4.2 Reinforcement learning on Atari tasks

We now evaluate the performance of AGS-CL on Atari [6] reinforcement learning (RL) tasks. As mentioned in the Introduction, a few previous works [13, 31] also considered the continual learning

of Atari tasks, but their settings allowed the agent to learn past tasks again in a *recurring* fashion. In contrast, we consider *pure* continual learning setting, namely, the past tasks cannot be learned again, but the average rewards are evaluated for all tasks learned so far after learning each task. We randomly selected eight Atari tasks, *i.e.*, {*StarGunner - Boxing - VideoPinball - Crazyclimber - Gopher - Robotank - DemonAttack - NameThisGame*}, and compared AGS-CL with three baselines, EWC, MAS and fine-tuning. The CNN agent had three convolution layers, one fully connected layer, and 8 separate output layer for each task, and we used PPO [30] identically for learning the agent for all comparing methods. Each task is learned with $10^7$ steps, and we evaluated the reward of the agent 30 times per task. We did fair hyperparameter search for all methods and report the best result for each method. More detailed experimental settings and are given in Supplementary Materials.

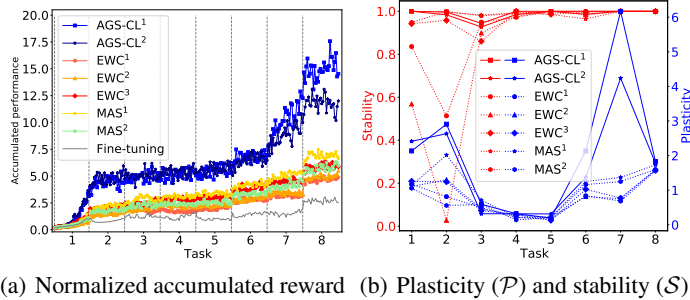

(a) Normalized accumulated reward  (b) Plasticity ($\mathcal{P}$) and stability ($\mathcal{S}$)

Figure 7: Reinforcement learning results. $\lambda = \{1, 2.5, 10\} \times 10^4$ for EWC[1,2,3], $\lambda = \{1, 10\}$ for MAS[1,2], and $\mu = 0.1$, $\lambda = \{1, 10\} \times 10^2$ for AGS-CL[1,2] were used, respectively.

Figure 7(a) shows the normalized accumulated rewards, in which each evaluated reward is normalized with the maximum reward obtained by fine-tuning for each task, for 8 tasks obtained by the baselines and AGS-CL. We clearly observe that AGS-CL achieves much superior accumulated reward at the end of the 8 tasks compared to both EWC and MAS ($\sim 3\times$) and fine-tuning ($\sim 5\times$). Figure 7(b) further considers the plasticity and stability for each task (instead of the average values as in supervised learning). We note AGS-CL hardly suffers from catastrophic forgetting (*i.e.*, stability $\approx 1$ for most tasks) and also does a much better job in learning new tasks than not only the EWC and MAS, but also the fine-tuning (*i.e*, plasticity $\gg 1$, particularly for tasks 1,2 and 7).

## 5  Concluding Remark

We proposed AGS-CL, a new continual learning method based on node-wise importance regularization. With a novel loss function based on group-sparsity norms, PGD optimization technique, and the re-initialization tricks, we showed our AGS-CL dominated other state-of-the-arts on various benchmark datasets even with *orders of magnitude* smaller number of regularization parameters. We also show the promixing results on *pure* continual reinforcement learning with Atari tasks.Our method can be also naturally extended to dynamic architecture-based method by simply adding some more free nodes to $\mathcal{G}_0^t$ when the network capacity depletes, which we will pursue as a future work.

## 6  Broader Impact

We tackle a fairly general continual learning problem, and there is no particular application forseen. The potential societal impact of our work, however, lies in saving intensive usage of computing resources, which is known to affect climate change and global warming due to the excessive energy consumption and necessity of cooling systems. Namely, when numerous ML applications require repetitive re-training of computationally intensive neural networks for learning every new task, overloading of data centers is indispensable. Hence, an effective continual learning algorithm, as proposed in our paper, can save such heavy energy consumption without losing the model accuracy. Furthermore, the effective memory usage can be additional benefit for using our method in memory-limited environments, *e.g.*, mobile devices.

## Acknowledgements

This work is supported in part by Institute of Information & communications Technology Planning Evaluation (IITP) grant funded by the Korea government (MSIT) [No.2016-0-00563, Research on adaptive machine learning technology development for intelligent autonomous digital companion], [No.2019-0-00421, AI Graduate School Support Program (Sungkyunkwan University)], [No.2019-0-01396, Development of framework for analyzing, detecting, mitigating of bias in AI model and training data], and [IITP-2019-2018-0-01798, ITRC Support Program].

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
