[Supplementary Material]

# Supplementary Materials for Continual Learning with Node-Importance based Adaptive Group Sparse Regularization

**Sangwon Jung**[1,*] **Hongjoon Ahn**[2,*] **Sungmin Cha**[1] **and Taesup Moon**[1,2]
[1]Department of Electrical and Computer Engineering, [2] Department of Artificial Intelligence,
Sungkyunkwan University, Suwon, Korea 16419
{s.jung, hong0805, csm9493, tsmoon}@skku.edu

## 1 Proof of Lemma 1

From (Eq.(3), manuscript), $\mathbf{prox}_{\alpha f}(\boldsymbol{v})$ minimizes the convex function

$$\ell(\boldsymbol{\theta}) \triangleq c\|\boldsymbol{\theta} - \boldsymbol{\theta}_0\|_2 + \frac{1}{2\alpha}\|\boldsymbol{\theta} - \boldsymbol{v}\|_2^2, \tag{1}$$

and for brevity, denote $\boldsymbol{\theta}^* := \mathbf{prox}_{\alpha f}(\boldsymbol{v})$ as the minimizer. Denoting $\partial_{\boldsymbol{\theta}}\ell(\boldsymbol{\theta})$ as the set of subgradients of $\ell(\boldsymbol{\theta})$, we know that $\boldsymbol{\theta}^* \in \{\boldsymbol{\theta} : \partial_{\boldsymbol{\theta}}\ell(\boldsymbol{\theta}) = 0\}$ since $\ell(\boldsymbol{\theta})$ is convex. Also, by denoting $\boldsymbol{w}$ as the subgradient of $\|\boldsymbol{\theta} - \boldsymbol{\theta}_0\|_2$ at $\boldsymbol{\theta}^*$, we then have the optimality condition,

$$\frac{1}{\alpha}(\boldsymbol{v} - \boldsymbol{\theta}^*) = c\boldsymbol{w}. \tag{2}$$

Since $\|\boldsymbol{\theta} - \boldsymbol{\theta}_0\|_2$ is not differentiable at $\boldsymbol{\theta} = \boldsymbol{\theta}_0$, we know

$$\boldsymbol{w} = \begin{cases} \frac{\boldsymbol{\theta}^* - \boldsymbol{\theta}_0}{\|\boldsymbol{\theta}^* - \boldsymbol{\theta}_0\|_2} & \text{if } \boldsymbol{\theta}^* \neq \boldsymbol{\theta}_0 \\ \in \{\boldsymbol{w} : \|\boldsymbol{w}\|_2 < 1\} & \text{if } \boldsymbol{\theta}^* = \boldsymbol{\theta}_0 \end{cases}. \tag{3}$$

Now, taking $\ell_2$-norm on both sides of (2), we can deduce

$$\boldsymbol{\theta}^* = \boldsymbol{\theta}_0 \text{ if and only if } \|\boldsymbol{v} - \boldsymbol{\theta}^*\|_2 < \alpha c. \tag{4}$$

Moreover, if $\boldsymbol{\theta}^* \neq \boldsymbol{\theta}_0$, we can derive from (2) and (3) that

$$\|\boldsymbol{v} - \boldsymbol{\theta}_0\|_2 - \alpha c = \|\boldsymbol{\theta}^* - \boldsymbol{\theta}\|_2 \geq 0, \tag{5}$$

and correspondingly,

$$\boldsymbol{\theta}^* = \left(1 - \frac{\alpha c}{\|\boldsymbol{\theta}_0 - \boldsymbol{v}\|_2}\right)\boldsymbol{v} + \frac{\alpha c}{\|\boldsymbol{\theta}_0 - \boldsymbol{v}\|_2}\boldsymbol{\theta}_0. \tag{6}$$

Combining (4) and (6), we have the lemma. ∎

## 2 Additional ablation studies

### 2.1 Ablation study of $\rho$

Here, we analyze the effect of $\rho$ for the **[Rand-init]** described in Section 3.4 (manuscript) (I.2). Figure 1 below reports the average accuracy on CIFAR-100 for AGS-CL and MAS. For AGS-CL, we fixed $(\mu, \lambda) = (10, 400)$ and varied $\rho \in \{0.1, \ldots, 0.5\}$, and for MAS, we used the optimal hyperparameter.

---

First, we observe that for $\rho \le 0.5$, AGS-CL is not very sensitive to $\rho$, and it outperforms MAS for all $\rho$. Second, we observe that $\rho$ affects the plasticity for learning new tasks. Namely, while $\rho = 0.1$ and $\rho = 0.5$ achieve the same final average accuracy, we note $\rho = 0.1$ suffers earlier since it does not sufficiently grow the network capacity for learning new tasks, whereas $\rho = 0.5$ suffers later since it uses up the network capacity too much in early tasks and makes the network too stable for later tasks. Thus, appropriate $\rho$ may find the right trade-off between the sparsity and the used capacity of the network and achieve higher average accuracy.

Figure 1: Average accuracy of AGS-CL on CIFAR-100 depending on $\rho$

## 2.2 Effect of PGD updates

(a) Average accuracy with and without PGD.

(b) Sparsity (decreasing curves) and used capacity (increasing curves) with and without PGD.

Figure 2: Ablation study on PGD for CIFAR-100

As mentioned in Section 3.3 (manuscript), our PGD update plays a critical role in achieving high accuracy. Here, we compare with a method without PGD. Figure 2(a) and Figure 2(b) show the average accuracy and the sparsity and used capacity on CIFAR-100. '*w/o* PGD' in Figure 2 indicates training the network without PGD, *i.e.*, the Adam step was used for optimizing $\mathcal{L}_t(\boldsymbol{\theta})$ (Eq.(2), manuscript) which implies the combined loss of $\mathcal{L}_{\mathrm{TS},t}(\boldsymbol{\theta})$ and group sparse regularizations(term (a) and term (b) of Eq.(2), manuscript). Since optimizing $\mathcal{L}_t(\boldsymbol{\theta})$ using Adam cannot achieve the global optimal point of group sparse regularization, we used a proper threshold $\tau$ to modify the definition of $\mathcal{G}_0$ in (Eq.(1), manuscript) and the used capacity. Thus, we define $\mathcal{G}_0^{t-1} \triangleq \{n_\ell : \Omega_{n_\ell}^{t-1} < \tau\} \subseteq \mathcal{G}$, and used capacity as $|\{n_\ell : \|\hat{\boldsymbol{\theta}}_{n_\ell}^{(t)} - \hat{\boldsymbol{\theta}}_{n_\ell}^{(t-1)}\|_2 < \tau\}|/|\mathcal{G}|$. Except for above definitions, all the common hyperparameters and training settings are same as '*w/* PGD', and we set the threshold $\tau = 10^{-4}$.

Followings are our observations. First, the average accuracy (Figure 2(a)) of '*w/o* PGD' is much lower than '*w/* PGD', which indicates that our PGD updates not only require *less* hyperparameters (*i.e.*, does not need $\tau$ threshold), but also does a much more accurate sparsification and freezing for achieving high accuracy. Second, we observe the sparsity (Figure 2(b)) of '*w/o* PGD' decreases

much faster than 'w/ PGD'. The reason is because the weights associated with the nodes in $\mathcal{G}_0^t$ are not exactly zero, hence, the gradients for those weights do not vanish, which cause the unimportant nodes in $\mathcal{G}_0^t$ also continuously learn in every task. From these results, we conclude our PGD update is essential in AGS-CL.

## 2.3 Comparison with EWC

We additionally evaluate the performance of EWC with two measures, plasticity ($\mathcal{P}$) and stability ($\mathcal{S}$), which are proposed in (Figure 5(c), manuscript). Figure 3 reports the trade-offs between $\mathcal{P}$ and $\mathcal{S}$ for AGS-CL, MAS and EWC. The plotted trade-offs of EWC are over the $\lambda$ and the others are the same as (Figure 5(c), manuscript). Note that although EWC has comparable $\mathcal{P}$-$\mathcal{S}$ trade-offs with MAS, AGS-CL apparently has the better $\mathcal{P}$-$\mathcal{S}$ trade-offs than EWC and MAS.

Figure 3: Plasticity ($\mathcal{P}$) and stability ($\mathcal{S}$) for CIFAR-100

# 3 Implementation details

## 3.1 Supervised learning

In CIFAR-100, CIFAR-10/100 and Omniglot [2], we train all methods with mini-batch size of 256 for 100 epochs using Adam optimizer [1] with initial learning rate 0.001 and decaying it by a factor of 3 if there is no improvement in the validation loss for 5 consecutive epochs, similarly as in [4]. In CUB200[3], we train all methods with mini-batch size 64 for 40 epochs using SGD with momentum 0.9 with initial learning rate 0.005 and decay it by a factor of 10 after training 30 epochs.

### 3.1.1 Hyperparameters for supervised learning experiments

The details on hyperparameters are in Table 1. For AGS-CL, we set $\eta$ to 0.9 and for RWALK, we set $\alpha$ to 0.9 for all datasets. We extensively searched the best hyperparameter for each method to make the comparison as fair as possible.

Table 1: Hyperparameters for supervised learning experiments

| Methods\Dataset | CIFAR-100 | CIFAR-10/100 | Omniglot | CUB200 | Sequence of 8 different datasets |
|---|---|---|---|---|---|
| AGS-CL | $\lambda$ (400) $\mu(10), \rho(0.3)$ | $\lambda$ (7000) $\mu(20), \rho(0.2)$ | $\lambda$ (1000) $\mu(7), \rho(0.5)$ | $\lambda$ (1.5) $\mu(0.5), \rho(0.1)$ | $\lambda$ (400000) $\mu(40), \rho(0.4)$ |
| EWC | $\lambda$ (10000) | $\lambda$ (25000) | $\lambda$ (500000) | $\lambda$ (40) | $\lambda$ (1000) |
| SI | c (1.0) | c (0.7) | c (0.85) | c (0.75) | - |
| RWALK | $\lambda$ (8) | $\lambda$ (6) | $\lambda$ (70) | $\lambda$ (50) | - |
| MAS | $\lambda$ (4) | $\lambda$ (1) | $\lambda$ (7) | $\lambda$ (0.6) | $\lambda$ (0.1) |
| HAT | c (2.5), smax(400) | c (0.1), smax(400) | c (2.5), smax(400) | - | - |

### 3.1.2 Details on network architectures

The details on network architectures for CIFAR-100, CIFAR-10/100 and Omniglot are in Table 2 and 3. Since the number of classes for each task is different in Omniglot, we denoted the classes of $i$th task as $C_i$. For CUB200, we use the AlexNet architecture from PyTorch official models. [4]. For the sequence of 8 different datasets, we use the model of which the size of kernel is changed to $3 \times 3$ and the rest is the same as AlexNet.

Table 2: Network architecture for CIFAR-100 and CIFAR-10/100

| Layer | Channel | Kernel | Stride | Padding | Dropout |
|---|---|---|---|---|---|
| 32×32 input | 3 | | | | |
| Conv 1 | 32 | 3×3 | 1 | 1 | |
| Conv 2 | 32 | 3×3 | 1 | 1 | |
| MaxPool | | | 2 | 0 | 0.25 |
| Conv 3 | 64 | 3×3 | 1 | 1 | |
| Conv 4 | 64 | 3×3 | 1 | 1 | |
| MaxPool | | | 2 | 0 | 0.25 |
| Conv 5 | 128 | 3×3 | 1 | 1 | |
| Conv 6 | 128 | 3×3 | 1 | 1 | |
| MaxPool | | | 2 | 1 | 0.25 |
| Dense 1 | 256 | | | | |
| Task 1 : Dense 10 | | | | | |
| . . . | | | | | |
| Task $i$ : Dense 10 | | | | | |

Table 3: Network architecture for Omniglot

| Layer | Channel | Kernel | Stride | Padding | Dropout |
|---|---|---|---|---|---|
| 28×28 input | 1 | | | | |
| Conv 1 | 64 | 3×3 | 1 | 0 | |
| Conv 2 | 64 | 3×3 | 1 | 0 | |
| MaxPool | | | 2 | 0 | 0 |
| Conv 3 | 64 | 3×3 | 1 | 0 | |
| Conv 4 | 64 | 3×3 | 1 | 0 | |
| MaxPool | | | 2 | 0 | 0 |
| Task 1 : Dense $C_1$ | | | | | |
| . . . | | | | | |
| Task $i$ : Dense $C_i$ | | | | | |

### 3.1.3 Result tables

Table 4 shows the detailed results used to generate (Figure 4, manuscript). The number in the paranthesis with $\pm$ sign stands for the standard deviation of the accuracy obtained from 5 independent runs with different random seeds.

## 3.2 Reinforcement learning

### 3.2.1 Details on network architectures

For training Atari 8 tasks, we used the same architecture which was proposed in [2]. However, to secure the model capacity for training 8 tasks well enough, we implemented each layer that has four times more filters than the original architecture. Figure 5 shows the details of our model.

### 3.2.2 Hyperparameters of PPO

We used PPO [3] as an algorithm for training Atari 8 tasks. Figure 6 shows hyperparameters that we used for 8 tasks, and these hyperparameters are equally applied to each baseline. We evaluate each method every 40 updates, *i.e.* we have 30 evaluation results during training each task. We trained the model using Adam optimizer with the initial learning rate of 0.0003 and the other hyperparameters are same as [3].

Table 4: Average accuracy(%) and standard deviation for 5 random seeds

|  | AGS-CL | EWC | SI | RWLAK | MAS | HAT |
|---|---|---|---|---|---|---|
| CIFAR-100 | **64.1** ($\pm$1.7) | 60.2 ($\pm$1.1) | 60.3 ($\pm$1.3) | 58.1 ($\pm$1.7) | 61.5 ($\pm$0.9) | 59.2 ($\pm$0.7) |
| CIFAR-10/100 | **76.1** ($\pm$0.4) | 70.0 ($\pm$0.3) | 71.5 ($\pm$0.5) | 69.6 ($\pm$1.1) | 72.1 ($\pm$0.7) | 59.8 ($\pm$1.6) |
| Omniglot | **82.8** ($\pm$1.8) | 76.0 ($\pm$20.2) | 54.9 ($\pm$16.2) | 71.0 ($\pm$5.6) | 81.4 ($\pm$2.1) | 5.5 ($\pm$11.1) |
| CUB200 | **81.9** ($\pm$0.7) | 80.5 ($\pm$1.2) | 80.4 ($\pm$0.8) | 81.0 ($\pm$1.3) | 79.6 ($\pm$1.0) | - |
| Sequence of 8 different datasets | **57.7** ($\pm$0.7) | 52.2 ($\pm$2.9) | - | - | 41.5 ($\pm$4.2) | - |

Table 5: Network architecture for Atari

| Layer | Channel | Kernel | Stride | Padding | Dropout |
|---|---|---|---|---|---|
| 84$\times$84 input | 4 |  |  |  |  |
| Conv 1 | 32$\times$4 | 8$\times$8 | 4 | 0 |  |
| ReLU |  |  |  |  |  |
| Conv 2 | 32$\times$4 | 4$\times$4 | 2 | 0 |  |
| ReLU |  |  |  |  |  |
| Conv 2 | 64$\times$4 | 3$\times$3 | 1 | 0 |  |
| ReLU |  |  |  |  |  |
| Flatten |  |  |  |  |  |
| Linear1 | 32$\times$4$\times$7$\times$7 |  |  |  |  |
| Task 1 : Dense $C_1$ |  |  |  |  |  |
| $\cdots$ |  |  |  |  |  |
| Task $i$ : Dense $C_i$ |  |  |  |  |  |

### 3.2.3 Detailed experimental results with $\mu = 0.1$

Figure 4: Reinforcement learning results. $\lambda = \{1, 2.5, 10\} \times 10^4$ for EWC[1,2,3], $\lambda = \{1, 10\}$ for MAS[1,2], and $\mu = 0.1$, $\lambda = \{1, 10\} \times 10^2$ for AGS-CL[1,2] were used, respectively.

Figure 4 shows detailed rewards during training each task. From this figure, we can clearly observe that AGS-CL outperforms EWC for Task 1, 2 and 7 significantly. Especially, for Task 7, AGS-CL showed higher rewards than Fine-tuning, which means it achieves significantly higher plasticity. We also note that AGS-CL has higher stability than other baselines for all $\lambda$.

### 3.2.4 Additional experimental results with $\mu = 0.125$

To show the other result with a different $\mu$, we selected $\mu = 0.125$ and experimented in Atari 8 tasks. From Figure 5, we observed that AGS-CL also achieves the highest reward , which is proposed in the manuscript, using $\mu = 0.1$ if we set an appropriate $\lambda$ for AGS-CL. Figure 6 shows detailed experimental results with $\mu = 0.125$. There is a little difference with the reward of each task in Figure 4 but we observed that AGS-CL shows similar advantages which we already mentioned in Section 3.2.3.

Table 6: Details on hyperparameters of PPO.

| Hyperparameters | Value |
|---|---|
| # of steps of each task | $10^7$ |
| # of processes | 128 |
| # of steps per iteration | 64 |
| PPO epochs | 10 |
| entropy coefficient | 0 |
| value loss coefficient | 0.5 |
| $\gamma$ for accumulated rewards | 0.99 |
| $\lambda$ for GAE | 0.95 |
| mini-batch size | 64 |

Figure 5: Normalized accumulated rewards. $\lambda = \{1, 2.5, 10\} \times 10^4$ for EWC[1,2,3], $\lambda = \{1, 10\}$ for MAS[1,2], and $\mu = 0.125$, $\lambda = \{1, 10\} \times 10^2$ for AGS-CL[1,2] were used, respectively.

Figure 6: Reinforcement learning results. $\lambda = \{1, 2.5, 10\} \times 10^4$ for EWC[1,2,3], $\lambda = \{1, 10\}$ for MAS[1,2], and $\mu = 0.125$, $\lambda = \{1, 10\} \times 10^2$ for AGS-CL[1,2] were used, respectively.

## Footnotes

[2]https://drive.google.com/file/d/1WxFZQyt3v7QRHwxFbdb1KO02XWLT0R9z/view?usp=sharing

[3]https://github.com/visipedia/tf_classification/wiki/CUB-200-Image-Classification

[4]https://github.com/pytorch/vision/blob/master/torchvision/models/alexnet.py