[Reviews · NeurIPS 2020]

Review 1

Summary and Contributions: This paper proposed a regularization method under the umbrella of continual learning by introducing group L1 norm on every neuron. The important neurons are frozen, at the same time, the unimportant neuron are forced to be sparse.

Strengths: This is an interesting topic. There are two sources for the catastrophic forgetting: model drift and negative transfer. Each source is restricted by a penalty item.

Weaknesses: The contribution is quite incremental. It is somehow a combination of pruning and task-specific neuron regularization. The assumption is task bounded. The architecture is Alex network, so it is not essential to apply continual learning, since the retraining is not expensive. The expression in proximal gradient is a little bit confusing. eq 5 holds for each layer rather than for whole loss. The reinitialization trick is just an intuition, lacking of firm theoretical explanation.

Correctness: yes

Clarity: It is well written, and pleasant to read.

Relation to Prior Work: yes

Reproducibility: Yes

Additional Feedback: It could be more convincing by showing results on larger datasets and neural networks.


Review 2

Summary and Contributions: This paper proposes two sparsity-based regularisation terms for continual learning which are induced by importance of nodes instead of weights and hence reduces memory cost significantly. The two terms can explicitly control the stability and plasticity of the model by splitting the hidden nodes into two groups: unimportant and important nodes. Proximal gradient descent is utilised to obtain the analytic form of the optimal solution of the regularisation terms. The learning process also includes a parameter update strategy for specifically alleviating forgetting and negative transfer.

Strengths: The motivation is sound and attractive. The proposed method is novel, sophisticated, and efficient in terms of memory cost. It provides a feasible way to combine model compression and continual learning in the fixed model capacity scenario, which could be an important contribution to the community.

Weaknesses: This paper in general is strong.

Correctness: No obvious problem in my understanding.

Clarity: This paper is generally well structured, however, the clarity could be improved. 1. An incoming weight of an important node could be an outgoing weight of an unimportant node, how does the mechanism update such a weight? Will it be frozen or nullified? Could the authors give some analysis about such a situation? 2. Since the PGD update is applied per epoch, does the number of epochs affect the performance significantly? Can this method be applied in online continual learning scenario, i.e. only 1 epoch is allowed? 3. Why only EWC is compared in experiments of reinforcement learning?

Relation to Prior Work: Relation to prior work has been clearly discussed in Sec.1.

Reproducibility: Yes

Additional Feedback:


Review 3

Summary and Contributions: The paper addresses the catastrophic forgetting problem in continual learning with a regularization-based method. It takes inspiration from model compression and focuses on neural network weights at node level (grouping weights responsible for a single activation). Contributions: The authors 1) use a node-based version of 2 regularizers - 1 inducing sparsity and 1 reducing change of weights, for nodes which were important for previous tasks. 2) use a simple feed-forward method to evaluate the importance of a node, which appears to be made possible by their sparsity regulariser. 3) use proximal gradient descent to reduce the number of hyperparameters. 4) explicitly trim connections to prevent negative transfer (degrading the performance of a previous task), and randomly initialise unimportant weights in order to increase the model’s capacity. 5) show that their method outperforms competing methods using significantly less memory.

Strengths: - I think the ideas of contributions 1), 2) and 4) are novel and potentially interesting for the community. - Overall, the improvement in performance with less memory is an important advancement.

Weaknesses: In my opinion, the evaluation setting could be improved. - The first set of tasks - “Supervised learning on vision datasets”, is dividing a dataset into multiple tasks. It would have been better if the tasks were from different datasets, so that the input domains are more different from each other, and thus fewer nodes could be reused. - The RL setting does not include all baselines from the first setting, and I couldn’t find an explanation why.

Correctness: I didn't find incorrect statement in the paper.

Clarity: I found most of the paper easy to follow. Here are a few points: - The use of proximal gradient descent isn’t motivated before section 3.3. - It might be good to add a 1-paragraph summary of the methods described in sections 3.2, 3.3, 3.4 in order to provide a better overview of your approach. -Question: In line 214 you describe the effects of Zero-init, Rand-init. It appears that, for an unimportant node n, all parameters which are multiplied by this node are set to 0 for future tasks. Therefore, this node shouldn’t be useable by the following layer in future tasks. Yet, the incoming weights to this node are randomly initialised, so its activation can be changed in the following tasks. Why randomly initialise weights for a node, which cannot be used later? I think I know what you meant to write, but I find the description confusing.

Relation to Prior Work: Related work is well outlined and the differences from the closest paper are sufficiently discussed.

Reproducibility: Yes

Additional Feedback:

[Author Response · NeurIPS 2020]



Figure 1: Results on 8 visually different tasks (left), comparison with Re-training (middle), and Atari RL (right).

**Mixture of different tasks [R4]** We carried out an additional continual learning experiment on eight tasks (as in [33, manuscript]) that consist of vision datasets with *different* domains:{CIFAR-10 / CIFAR-100 / MNIST / SVHN / Fashion-MNIST / Traffic-Signs / FaceScrub / NotMNIST}. Figure 1(left) compares the average accuracy of AGS-CL with two most stable baselines, EWC and MAS, and Fine-tuning. We clearly observe AGS-CL again significantly dominates EWC and MAS, confirming the effectiveness of our approach in a more challenging setting.

**Comparison with Re-training [R1]** For CUB200, we compared AGS-CL with Re-training, which takes a pre-trained AlexNet and re-trains for each task with the entire training sets observed so far. In terms of accuracy, AGS-CL (82.3%) was just slightly lower than Re-training (86.5%), which is an obvious upper bound. The clear benefit of AGS-CL, however, is shown in Figure 1(middle) in terms of the training time per epoch (red) and memory requirement (blue). We note both AGS-CL and Re-training had 40 training epochs for each task, and for the required memory, Re-training needs to store all the training data observed so far, whereas AGS-CL needs to store one additional AlexNet model and $\{\Omega_{n_\ell}^t\}_{n_\ell \in \mathcal{G}}$. We can clearly observe from the figure that the training time and memory for AGS-CL remain constant, whereas those for Re-training grow linearly (*i.e.*, *very expensive*), as the number of tasks grows. We believe this result clearly responds to **[R1]** and justifies the necessity of continual learning with AlexNet.

**Re-initialization [R3,R4]** We stress that the **order** of the re-initializations are important, *i.e*, [Zero-init] is always followed by [Rand-init]. (We will re-emphasize this in the final version.) Also, note that the outgoing weights of an unimportant node can be connected to *either* important *or* unimportant nodes in the upper layer. In such a case, our [Zero-init] *nullifies* and *fixes* those connected to the important nodes (Re:**[R3]**), but [Rand-init] re-utilizes those connected to the unimportant nodes so that they can become learnable for the next task (Re:**[R4]**). This is also illustrated in the rightmost network figure in Figure 2 (manuscript); note the activation of the unimportant (gray) node in the second layer can be used for learning the unimportant (gray) node in the third layer for task $t + 1$ (via the gray, solid weight between them that is randomly re-initialized). We will make sure to give a more clarified explanation on the re-init schemes in our final version.

**RL [R3,R4]** In the RL setting, it is nontrivial to implement SI/RWALK, which require to compute the gradient path integrals, or HAT, which implements hard attention mask for each task. EWC is the only scheme among our baselines that had results for RL, with the limitation mentioned in line 60, and that was the reason why we originally only compared with it. Now, since MAS operates almost similarly as EWC (in terms of the learning process), we also implemented MAS for the Atari RL tasks, and Figure 1(right) shows the results (for a single random seed) together with AGS-CL, EWC, and Fine-tuning. We observe MAS performs almost the same as EWC, and again, AGS-CL convincingly outperforms both. We will add this result as well as the explanation on why we excluded other baselines in the final version.

**[R1]** ① We respectively disagree that our paper has only incremental contributions. As also clearly listed by **[R4]** and appreciated by **[R3]**, we believe our method combines several different principles in a novel way for continual learning. In our opinion, *"It is somehow . . . regularization."* is quite vague, which makes it hard to make a systematic rebuttal. ② Eq. (5) is a general expression that *defines* the proximal gradient descent. In AGS-CL, the proximal operator in Eq.(5) can be applied independently for each node, and Eq.(5) becomes Eq.(7). This is also clearly mentioned in line 170∼173, and we hope this helps to resolve the confusion. ③ Regarding the re-initialization, we cannot see why using an *intuition* would be problematic. Numerous algorithms for deep learning, *e.g.*, dropout, batch normalization, attention mechanism, or weight initialization, are based on sound intuitions with little theoretical explanation. They are shown to be very effective, typically via extensive experiments, which we believe is also done in our paper as well (*e.g.*, Figure 5(b)(c)).

**[R3]** ① Our method can be also applied to the online continual learning setting. However, since the PGD leads to the sparsification or freezing of weights after a couple of epochs, the performance could be limited for the 1 epoch setting.

**[R4]** ① We mentioned in line 45∼46 that PGD was used as a tool to elegantly optimize our loss function and is described in details in Section 3.3. We will make sure to more clearly motivate PGD in the final version. ② Thanks for the comments. We will write a clear 1-paragraph summary to provide readers with a better overview of our method.

[Meta-Review · NeurIPS 2020]

The paper provides a different way of thinking about regularizing neural networks in continual learning. Their idea stems from network compression. The authors have explained the motivation and the method is fairly justified, and none of the prior work has done it in a similar way as proposed here. The authors also addressed most of the issues raised by the reviewers in their rebuttal.